# Sex-Based Disparities in Leukocyte Migration and Activation in Response to Inhalation Lung Injury: Role of SDF-1/CXCR4 Signaling

**DOI:** 10.3390/cells12131719

**Published:** 2023-06-26

**Authors:** Tanima Chatterjee, Terry L. Lewis, Itika Arora, Anastasiia E. Gryshyna, Lilly Underwood, Juan Xavier Masjoan Juncos, Saurabh Aggarwal

**Affiliations:** Division of Molecular and Translational Biomedicine, Department of Anesthesiology and Perioperative Medicine, School of Medicine, University of Alabama at Birmingham, Birmingham, AL 35205-3703, USA; tchatterjee@uabmc.edu (T.C.); terrylewis@uabmc.edu (T.L.L.); itika.arora@cchmc.org (I.A.); gryshyna@uab.edu (A.E.G.); lillyunderwood@uabmc.edu (L.U.); jxjuncos@uabmc.edu (J.X.M.J.)

**Keywords:** chlorine, gender, lung injury, leukocyte

## Abstract

The aim of the study was to determine whether sex-related differences exist in immune response to inhalation lung injury. C57BL/6 mice were exposed to Cl_2_ gas (500 ppm for 15, 20, or 30 min). Results showed that male mice have higher rates of mortality and lung injury than females. The binding of the chemokine ligand C-X-C motif chemokine 12 (CXCL12), also called stromal-derived-factor-1 (SDF-1), to the C-X-C chemokine receptor type 4 (CXCR4) on lung cells promotes the migration of leukocytes from circulation to lungs. Therefore, the hypothesis was that elevated SDF-1/CXCR4 signaling mediates exaggerated immune response in males. Plasma, blood leukocytes, and lung cells were collected from mice post-Cl_2_ exposure. Plasma levels of SDF-1 and peripheral levels of CXCR4 in lung cells were higher in male vs. female mice post-Cl_2_ exposure. Myeloperoxidase (MPO) and elastase activity was significantly increased in leukocytes of male mice exposed to Cl_2_. Lung cells were then ex vivo treated with SDF-1 (100 ng/mL) in the presence or absence of the CXCR4 inhibitor, AMD3100 (100 nM). SDF-1 significantly increased migration, MPO, and elastase activity in cells obtained from male vs. female mice post-Cl_2_ exposure. AMD3100 attenuated these effects, suggesting that differential SDF-1/CXCR4 signaling may be responsible for sex-based disparities in the immune response to inhalation lung injury.

## 1. Introduction

There is a general consensus that sex is an important biologic variable in immunity. For instance, autoimmune diseases are more prevalent in females [1], while males have a higher risk of malignancy and mortality associated with malignancy [2,3]. Males are generally more susceptible than females to infections caused by viruses, bacteria, parasites, and fungi [4]. Females mount higher antibody responses to bacterial and viral vaccines compared to their male counterparts [5]. Animal studies report that testosterone-mediated suppression of immunity is responsible for higher mortality in male vs. female C57BL/6 mice infected with *Plasmodium chabaudi* [6]. In fact, the most recent pandemic also exposed major differences in male vs. female immune responses to COVID-19 infection. COVID-19 symptoms were reported to be more severe and mortality was higher in males [7]. However, it is still not known whether sex-based differences exist in the immune response to inhalation lung injury caused by exposure to chlorine (Cl_2_) gas.

In 2016, the American Association of Poison Control Centers reported over 6300 exposures to Cl_2_, making it the most common inhalational irritant in the United States [8]. However, current treatment strategies are insufficient and merely palliative. Understanding the pathogenesis and any sex-based differences in response to Cl_2_ inhalation will help improve management strategies. Acute and chronic lung injury post-Cl_2_ exposure is due to an uncontrolled activation of alveolar macrophages, and sequestered macrophages and neutrophils in lungs [9,10,11,12,13,14,15]. Excessive recruitment of leukocytes is critical to the pathogenesis of lung injury, and the magnitude and duration of the inflammatory process may ultimately determine the outcome in patients [16,17].

The first step in the pathogenesis of inflammation is transmigration of circulating leukocytes through the activated vascular endothelial cell in the inflamed tissue. During this process, leukocytes are activated to secrete a variety of substances such as growth factors, chemokines and cytokines, proteases, nitric oxide, and reactive oxygen metabolites, which are considered to be one of the primary sources of tissue injury. Prevention of or reduction in leukocyte migration can profoundly attenuate the parenchymal cell dysfunction during inflammation. Under homeostatic conditions, stromal-derived-factor-1 (SDF-1) and its cognate receptor, C-X-C chemokine receptor type 4 (CXCR4) are critical for hemopoietic development and homing of neutrophils and lymphocytes in bone marrow [18,19,20,21]. However, under conditions of stress, elevated SDF-1/CXCR4 signaling in injured tissue promotes the release of neutrophils from bone marrow into blood [22,23,24] and the subsequent migration and homing of the circulating neutrophils to the site of injury. Furthermore, ex vivo studies reveal that SDF-1/CXCR4 signaling acts not only as a chemoattractant, but also as a suppressor of cell death for the lung neutrophils expressing CXCR4 [25]. The SDF-1/CXCR4 axis has also been demonstrated to be responsible for the tissue migration of neutrophils and promoting inflammation in animal models of arthritis [26,27,28,29] and peritonitis [30]. In this study, we investigated whether SDF-1/CXCR4 signaling plays a role in sex-based differential immune response to lung injury post-Cl_2_ exposure.

## 2. Methods

### 2.1. Mouse Model

Adult male and female C57BL/6 mice (20–25 g) were purchased from Jackson labs (Farmington, CT, USA). All mice were housed under a 12 h light/12 h dark cycle with ad libitum access to standard diet and water. Euthanasia protocol based on intraperitoneal injections of ketamine and xylazine (100 and 10 mg/kg body weight, respectively) was used in the study for mice to minimize pain and distress. All animal care and experimental procedures were approved by the Institutional Animal Care and Use Committee at UAB (Animal Protocol number: IACUC-22333).

### 2.2. Cl_2_ Exposure

Mice were exposed to air or Cl_2_ (500 ppm) for 15, 20, or 30 min in environmental chambers and then returned to room air. Mice were euthanized at different intervals (1, 7, or 14 day) post-air or -Cl_2_ exposure. Before exposure to Cl_2_, mice received a subcutaneous injection of buprenorphine SR (1 mg/kg) to minimize pain.

### 2.3. Isolation of Bronchoalveolar Lavage Fluid (BALF) and Blood Leukocytes

Bronchoalveolar lavage was performed on whole lung and the recovered lavage fluid was centrifuged immediately at 3000× *g* for 10 min to pellet cells. BCA protein assay was performed on supernatant, while pelleted cells were counted using Neubauer hemocytometer. To isolate leukocytes, blood drawn from mice was mixed with an equal volume of 3% dextran for 30 min to separate leukocyte-rich plasma. Leukocyte containing supernatant was centrifuged (10 min, 1000 rpm, 4 °C) to collect pelleted RBCs and leukocytes. RBCs were lysed by adding hypotonic solution and leukocytes were re-suspended in 1× PBS [31].

### 2.4. Isolating Single Cell Suspension from Whole Lung

Mouse lungs were perfused with DPBS via the right ventricle. PBS-perfused lungs were isolated with other mediastinal organs. Dispase II solution was instilled into lungs through the trachea, which was ligated with a silk suture. After incubation at 37 °C for 50 min, lungs were separated from other mediastinal organs. Lungs were then minced and digested in 1× PBS with 0.1% collagenase, 0.01% deoxyribonuclease I, and 5 mM CaCl_2_ at 37 °C for 20 min. Dissociated lung cells were centrifuged at 2000 rpm for 10 min at 4 °C and the pellet was resuspended in cold 1× RBC lysis buffer to remove RBCs and subsequently washed with PBS. Cell viability was determined by staining with trypan blue.

### 2.5. SDF-1 Protein Quantification

Mouse plasma SDF-1 levels were quantified using the Quantikine mouse CXCL12/SDF-1α ELISA kit (Cat # MCX120; R&D systems, Minneapolis, MN, USA) according to manufacturer’s instructions.

### 2.6. Evaluation of Surface Levels of CXCR4 in Lung Cells Using Flow Cytometry

Peripheral levels of CXCR4 on isolated lung cells were determined by flow cytometry. Lung cell suspension was resuspended in FACS staining buffer and incubated on ice with Fc-γ receptor blocking solution for 15 min. Cells were washed by adding 800 µL of FACS buffer to cell suspension and centrifuged. Pellet was then incubated for 45 min at 4 °C with anti-CXCR4PE(Cat # 153806, BioLegend, San Diego, CA, USA) in combination with the viability dye, 7AAD (Cat # 420403, BioLegend). Cells were also incubated with the control isotype (Cat # 400507, BioLegend) corresponding to each primary antibody. Cells were then washed and passed through a 70 µm cell strainer into 5 mL polystyrene round-bottomed flow tube. Data were acquired by FACS Attune Flow Cytometer (BD Biosciences, Franklin Lakes, NJ, USA) and analyzed using FlowJo Software (Version 10.9, Ashland, OR, USA).

### 2.7. Leukocyte Migration Assay

The migration of immune cells was assessed by QCM Chemotaxis 3 µm Cell Migration assay (Cat # ECM505; Millipore Sigma, Burlington, ME, USA). Isolated lung cells were plated on a modified Boyden chamber with 3 μm pores to evaluate the migration upon stimulation by SDF-1. The 3 µm pore size of the permeable membrane is appropriate for leukocyte migration, however, other cells, such as epithelial and fibroblast cells, are too large (require 8 µm pore size), preventing their migration. A total of 20 × 10^4^ cells in 100 µL of RPMI 1640 medium containing 0.25% bovine serum albumin were placed in migration chambers. SDF-1 (100 ng/mL) (Cat # 460-SD, R&D systems, Minneapolis, MN, USA) was added to 150 µL serum-free media in the lower chamber, with some cells being pre-treated with AMD3100 (100 nM, 37 °C, 30 min) (Cat # 239825, Millipore Sigma, Billerica, MA, USA). The plate was subsequently incubated for 24 h at 37 °C. Cells in the lower chamber were collected and transferred to a new 96 well plate and the percent of migration determined from the original cell input according to manufacturer’s protocol.

### 2.8. Myeloperoxidase Activity Assay

Leukocyte myeloperoxidase (MPO) activity was assessed using Myeloperoxidase Activity assay kit (Cat # ab105136; abcam, Boston, MA, USA) in isolated blood leukocytes and lung cells. Lung cells were treated with SDF-1 (100 ng/mL) in the presence or absence of AMD3100 (100 nM) and leukocyte MPO activity was assessed 1-day post-treatment.

### 2.9. Elastase Activity Assay

Leukocyte elastase activity was assessed using Elastase assay kit (Cat# E12056; ThermoFisher Scientific, Waltham, MA, USA) in blood leukocytes and isolated lung cells. Lung cells were treated with SDF-1 (100 ng/mL) in the presence or absence of AMD3100 (100 nM) and leukocyte elastase activity was assessed 1-day post-treatment.

### 2.10. Statistics

Statistical analysis was performed using GraphPad Prism version 9.5 for Windows (GraphPad Software, San Diego, CA, USA). Results were expressed as mean ± standard error of the mean (SEM). Statistical significance was determined by unpaired *t*-test for two groups or two-way ANOVA followed by Tukey’s post-hoc test for more than two groups, *p* < 0.05 was considered significant.

## 3. Results

Figure 1 demonstrates that exposure to chlorine gas disproportionately increases mortality in male mice compared to female mice. The survival rate for adult male C57BL/6 mice (20–25 g body weight) is 0% within 1 day when they are exposed to Cl_2_ gas (500 ppm) for 30 min (Figure 1A) or 20 min (Figure 1B). In comparison, the survival rate for age-matched female mice is approximately 40% when exposed to Cl_2_ gas (500 ppm) for 30 min or 50% when they are exposed to Cl_2_ for 20 min 7 days post-exposure. Upon exposure to Cl_2_ gas for 15 min, both male and female mice have more than 70% survival rate over 14 days (Figure 1C). Therefore, for our future studies, we exposed mice to 15 min of Cl_2_ (500 ppm) gas. To determine whether lung inflammatory response to Cl_2_ exposure differs among male and female mice, lung injury parameters such as bronchoalveolar lavage fluid (BALF) protein levels and cell count were assessed in these mice (five male and five female). Data show that male mice have significantly higher BALF protein levels at 1- and 7- days post-Cl_2_ exposure, compared to female mice (Figure 2A). Similarly, analysis of total cell count shows that male mice have 4.2 × 10^6^, while female mice have 1.2 × 10^6^ cells in BALF 1-day post-Cl_2_ exposure. BALF cell count is also higher in male mice compared to female mice 7-days post-exposure (Figure 2B). Together, data show that male mice are more susceptible to Cl_2_-induced lung morbidity and mortality.

Next, to determine gender-based differences in SDF-1/CXCR4 signaling, we measured SDF-1 and CXCR4 levels in mice post-Cl_2_ exposure. Exposure to Cl_2_ gas increases the SDF-1 levels in BALF of male mice 1- and 7- days post-exposure and both male and female mice 14-days post-exposure (Figure 3A). Plasma SDF-1 levels are much higher in male mice at 1-day (three-fold) and 7-day (two-fold) post-exposure compared to female mice (Figure 3B). Next, lung cells were isolated from mice and peripheral levels of CXCR4 determined by FACS analysis (Figure 3C). Data show that CXCR4 levels significantly increase at 7- and 14- days post-Cl_2_ exposure in male mice (Figure 3D). However, the peripheral levels of CXCR4 decreases in female mice by day 14 post-Cl_2_ exposure. Together, data show that differential regulation of SDF-1/CXCR4 occurs in male vs. female mice post-Cl_2_-gas-induced inhalation lung injury.

To determine the role of SDF-1/CXCR4 signaling in immune cell migration, male and female mice were exposed to air or Cl_2_ gas (500 ppm, 15 min). Blood leukocytes and lung cells were isolated from mice 1- and 14- days post-exposure to air or Cl_2_. Immune cell migration was assessed by QCM Chemotaxis 3 µm cell migration assay. Cells were plated on a 3 µm permeable membrane in the upper chamber, which allowed leukocyte migration but prevented other cells, such as epithelial and fibroblast cells, due to size exclusion. Under basal conditions, migration of blood leukocytes across the permeable membrane is higher in male but not female mice 14 days post-Cl_2_-gas exposure (Figure 4A). On the contrary, there is no increase in the migration of lung leukocytes in male vs. female mice post-Cl_2_ exposure compared to air-exposed mice. However, lung leukocyte migration is significantly lower 14-days post-Cl_2_ exposure in female mice as opposed to male mice (Figure 4B). Then, some cells were pretreated with AMD3100 (100 nM), while SDF-1 (100 ng/mL) was added to media in the lower chamber overnight. Migration of leukocytes was measured next morning. In air-exposed animals, SDF-1 does not significantly increase migration of leukocytes (Figure 4C). However, data show that SDF-1 increases migration (three- fold) in lung leukocytes obtained from male mice 1-day post-Cl_2_ exposure compared to their female counterparts (Figure 4D), while AMD3100 attenuates the migration of cells. Similarly, SDF-1 also increases migration in leukocytes (two-fold) obtained from male mice 14-days post-Cl_2_ exposure (Figure 4E). However, there is no significant increase in migration in leukocytes obtained from female mice. Together, data show that SDF-1 promotes lung leukocyte migration post-Cl_2_-gas exposure in male but not female mice, due to higher levels of CXCR4 on lung cells. Whether CXCR4 levels are increased in blood leukocytes and are responsible for migration in response to SDF-1 will be the focus of future studies.

To determine whether SDF-1/CXCR4 signaling is also involved in leukocyte activation post-inhalation lung injury, male and female mice were exposed to air or Cl_2_ gas (500 ppm, 15 min). Leukocyte MPO activity was assessed in leukocytes isolated from blood and lung cells from air-exposed mice and from mice 1- and 14- days post-Cl_2_ (500 ppm, 15 min) exposure. Data show that leukocyte MPO activity is elevated in blood leukocytes (Figure 5A) and lung cells (Figure 5B) of male mice 1-day post-Cl_2_ exposure. Lung cells were then treated with SDF-1 (100 ng/mL) in the presence or absence of AMD3100 (100 nM) and leukocyte MPO activity assessed 1-day post-treatment. MPO activity is elevated in both male and female air-exposed mice post-SDF-1 treatment (Figure 5C). Leukocyte MPO activity also increases by more than 50% in SDF-1-treated lung cells obtained from male mice 1- (Figure 5D) and 14- days (Figure 5E) post-Cl_2_ exposure, which is attenuated by AMD3100. Together, leukocyte MPO activity results show that SDF-1/CXCR4 signaling is involved in lung leukocyte activity post-Cl_2_-gas exposure, mostly in male mice. The increase in blood leukocyte activity in male mice post-Cl_2_-gas exposure could be due to increased circulating levels of chloride ions post-Cl_2_-gas exposure. Chloride is a substrate for MPO, which converts hydrogen peroxide and chloride to hypochlorous acid. Whether SDF-1/CXCR4 signaling plays any significant role in blood leukocyte activation will be studied in the future.

Next, leukocyte elastase activity was assessed in blood leukocytes and isolated lung cells from air-exposed mice and cells obtained from mice exposed to Cl_2_ (500 ppm, 15 min), 1- and 14- days post-exposure. Data show that elastase activity is not changed in blood leukocytes post-Cl_2_ exposure (Figure 6A). However, leukocyte elastase activity increases by nearly five-fold in lung cells from male mice 1-day post-Cl_2_ exposure (Figure 6B). Lung cells were then treated with SDF-1 (100 ng/mL) in the presence or absence of AMD3100 (100 nM) and leukocyte elastase activity assessed 1-day post-treatment. SDF-1 does not increase elastase activity in lung cells of air-exposed mice (Figure 6C) but increases the enzymatic activity in cells isolated from male mice 1- (Figure 6D) and 14- (Figure 6E) days post-Cl_2_ exposure. AMD3100 attenuates this increase. These findings clearly show that SDF-1/CXCR4 signaling is mostly involved in regulating lung leukocyte activity in males post-Cl_2_-gas exposure.

## 4. Discussion

Males have larger lungs [32,33], however, the number of alveoli per unit area and per unit area volume and alveolar dimensions are similar in males and females [34]. Moreover, the intrinsic elasticity of lung parenchyma is similar between sexes [35]. Therefore, anatomical differences may not be responsible for how either male and female lungs respond to environmental insults. The aim of the study was to elucidate whether sex-related differences occur in response to inhalation lung injury caused by exposure to Cl_2_ gas. Our data show that male C57BL/6 mice are more susceptible to acute lung injury and have higher mortality post-Cl_2_-gas exposure in comparison to their age-matched female counterparts. We observe 0% survival within 24 h in male mice that are exposed to 500 ppm of Cl_2_ gas for either 30 or 20 min. In comparison, 40–50% of female mice survive post-exposure to 500 ppm of Cl_2_ for 30 or 20 min. In a previous study published by our partner group, there was no difference in the mortality rate between male and female mice upon Cl_2_ exposure [11]. Although these results seem contradictory, in the previous study, male and female mice were exposed to 600 ppm of Cl_2_ gas for 45 min. This high dose of Cl_2_ and longer duration of exposure resulted in 100% mortality within 24 h in mice of both sexes [11]. One goal of the current study was to examine lung injury following survivable Cl_2_ exposure. Therefore, data from these independent studies should not be compared. 

Our data further show that Cl_2_-inhalation-induced lung injury is more pronounced in male mice compared to their age-matched female counterparts. Male mice have higher levels of protein and inflammatory cells in BALF compared to females post-Cl_2_ exposure. Sex-specific differences are also evident in hyperoxia-induced lung injury [36]. After exposure to hyperoxia, males show greater lung injury, neutrophil infiltration, and apoptosis compared to females [36]. The difference in lung morbidity is due to higher levels of inflammation and oxidative stress in males [36]. In contrast, exposure to ozone results in greater lung injury in female compared to male mice [37]. Ozone exposure causes more exaggerated inflammation and oxidative damage in female mice compared to male mice [37]. An important factor differentiating hyperoxia- vs. ozone-induced lung injury is that the immune response to hyperoxia is mainly neutrophil infiltration into lungs [36], whereas ozone exposure mostly results in the accumulation of macrophages in the lower airways [37]. Therefore, it is possible that male mice experience severe inflammation post-Cl_2_ exposure because the majority of infiltrating cells are neutrophils rather than macrophages.

The recruitment of leukocytes to the sites of inflammation and leukocyte-derived inflammatory mediators contribute to the development of tissue injury associated with inflammation [16,17]. The SDF-1/CXCR4 axis promotes tissue migration of neutrophils and inflammation [26,27,28,29,30]. It was reported that SDF-1/CXCR4 signaling was involved in hyperimmune response and airway hyperreactivity in asthmatic animals [38,39]. SDF-1 was also up-regulated in an animal model of lung fibrosis [40,41]. Blocking CXCR4 with AMD3100 significantly attenuated fibrotic changes [42,43], likely due to the inhibition of fibrocyte mobilization to the injured lung. Reports also describe the role of SDF-1 in the homing of both malignant metastases and adult stem cells to the lungs in rodent models of metastatic cancers [44,45,46]. Using an anti-SDF-1 blocking antibody suppressed airspace neutrophilia in the lungs of animals exposed to lipopolysaccharide (LPS) [47]. In our study, SDF-1 and peripheral lung cell levels of CXCR4 are significantly higher post-Cl_2_ exposure in male mice compared to female mice. Therefore, there are clear sex differences in the regulation of SDF-1/CXCR4 signaling. One of the key regulators of SDF-1 and CXCR4 under hypoxic conditions is hypoxia-inducible factor-1 (HIF-1) [48,49,50]. Previously, it was reported that the administration of dihydrotestosterone (DHT) increased SDF-1 and CXCR4 levels in a mouse model of hindlimb ischemia [51]. As exposure to Cl_2_ gas is known to cause hypoxia [52], it is possible that the increase in SDF-1 and CXCR4 after Cl_2_ exposure is due to the upregulation of HIF-1. Interestingly, androgens are known to stimulate HIF-1 expression [53], therefore, it is likely that higher SDF-1/CXCR4 levels in male mice are due to androgen-dependent HIF-1 activation. Estrogen is also known to regulate SDF-1/CXCR4 levels [54], especially in hormone-dependent cancers, through estrogen receptor β (Erβ) [55]. The increased levels of SDF-1/CXCR4 promotes migration and metastasis of neoplastic cells [56]. Whether Cl_2_ exposure modulates the expression of Erβ in female mice is unknown and would be interesting to study in the future.

In our study, we find that ex vivo exposure to SDF-1 results in increased transmigration of leukocytes isolated from male but not female mice 1- and 14- days post-exposure to Cl_2_ gas (500 ppm, 15 min). This is inhibited by co-treatment of cells with the CXCR4 blocker AMD3100, suggesting that the overexpression of CXCR4 in the lungs of male mice is responsible for SDF-1-induced chemotaxis. However, it is worth mentioning that although SDF-1/CXCR4 signaling plays a significant role in neutrophil migration from the intravascular to the extravascular compartments of lungs, especially in the acute lung injury setting, other factors such as chemokines, cytokines, selectins, and other biomolecules may also be involved [57]. Several pro-inflammatory cytokines such as TNF-α, IL-1β, and G-CSF have been shown to induce leukocyte recruitment to lungs through upregulation of adhesion molecules [58,59]. The master inductor of these cytokines, nuclear factor (NF)-κB, is, therefore, also responsible for leukocyte migration [60,61]. Apart from SDF-1, other chemokines such as CXCL1, -2, and -5 are known to chemoattract leukocytes via binding to their receptor, CXCR2, as CXCR2-null mice do not exhibit LPS-induced alveolar neutrophilia [62]. In addition to cytokines and chemokines, matrikines, such as the tripeptides *N*-acetyl proline–glycine–proline (PGP), which are generated in airspace at later stages post-injury as a result of proteolytic activity of MMP8/9 and prolyl endopeptidase on collagen, are known to sustain leukocyte trafficking to lungs by binding to CXCR1/2 [63,64]. Therefore, it is very much possible that these matrikines may also be involved in late migration of leukocytes in Cl_2_-exposed male mice (14-days post-exposure). Whether gender differences occur in the generation of matrikines is still not known and subject of future studies. Finally, bioactive lipids such as leukotrienes (LTB4), which are generated from arachidonic acid (AA) and are known to be increased in lung injury, have also been shown to play a role in leukocyte trafficking to lungs [65,66,67]. In addition, LTB4 activates neutrophils that promote the release of lysosomal enzymes and reactive species production [68]. Clear sex differences in the production of pulmonary eicosanoids such as LTB4, PGE_2_, 6-keto-PGF_1α_, and 12S-hydroxyeicosateraenoic acid (HETE) have been associated with inflammation [69,70]. Compared with males, ozone-exposed female mice had elevated levels of PGE_2_, while the loss of ovarian hormones post-ovariectomy exacerbated pulmonary inflammation and injury [71]. Whether these biolipids are involved in sex-based differences in Cl_2_-induced lung injury is yet to be determined.

MPO and elastase are the major enzymes responsible for neutrophil-mediated tissue injury. MPO is highly abundant and comprises about 5% of the dry mass of azurophilic granules of neutrophils [72]. MPO catalyzes the formation of hypochlorous acid (HOCl) from hydrogen peroxide (H_2_O_2_) and chloride ions (Cl^−^) and also other highly reactive molecules, such as tyrosyl radicals [73]. The extracellular release of MPO, and subsequent production of these free radicals, promotes tissue injury [74,75]. Our results indicate that SDF-1 increases MPO and elastase activity ex vivo in lung cells isolated from mice post-Cl_2_-gas exposure. AMD3100 inhibits the activity of these enzymes, suggesting a clear role of CXCR4 in leukocyte activation. Our data also show that under basal conditions, there appears to be no difference in MPO activity between male and female leukocytes. However, there are conflicting reports in the literature, with Emokpae et al. showing that MPO activity is higher in females [76], while Gupta et al. show that the expression of MPO gene is higher in males [77].

## 5. Conclusions

In conclusion, this study outlines an animal model of inhalation lung injury that shows mechanisms regulating sex differences in leukocyte recruitment and activation that has important implications for disease management. Treatment for lung injury post-Cl_2_ exposure is mostly supportive, consisting of low tidal volume ventilation and fluid restriction. However, to further improve outcomes, specific gender-based targeted therapies such as CXCR4 inhibition, which limits inflammatory lung injury, may go a long way in improving morbidity.

## Figures and Tables

**Figure 1 cells-12-01719-f001:**
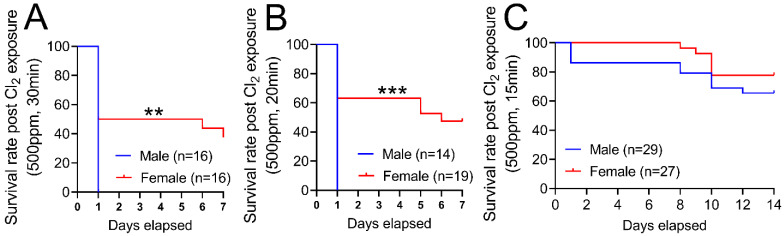
Cl_2_ exposure increases mortality in male vs. female mice. Kaplan–Meyer curves show that adult male C57BL/6 mice exposed to Cl_2_ gas (500 ppm) for 30 min (**A**) or 20 min (**B**) have 0% survival within 1-day. In comparison, adult female C57BL/6 mice have 40% and 50% survival rates 7-days post-exposure to Cl_2_ gas (500 ppm) for 30 (**A**) or 20 min (**B**), respectively. Upon exposure to Cl_2_ gas (500 ppm) for 15 min, both male and female mice have a more than 70 percent survival rate over 14-days (**C**). ** *p* < 0.01, *** *p* < 0.001 vs. male mice; log-rank (Mantel–Cox) test.

**Figure 2 cells-12-01719-f002:**
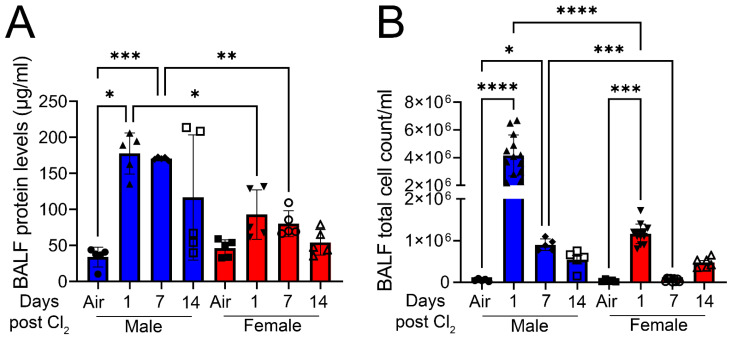
Cl_2_ increases BALF protein and total cell count levels primarily in male mice. Male C57BL/6 mice exposed to Cl_2_ gas (500 ppm, 15 min) have significantly elevated parameters of lung injury such as BALF protein levels (**A**) and cell count (**B**) compared to their female counterparts (n = 5–13). Individual values and means ± SEM. * *p* < 0.05, ** *p* < 0.01, *** *p* < 0.001, **** *p* < 0.0001 vs. groups at the end of individual lines; two-way ANOVA followed by Tukey’s post-hoc testing.

**Figure 3 cells-12-01719-f003:**
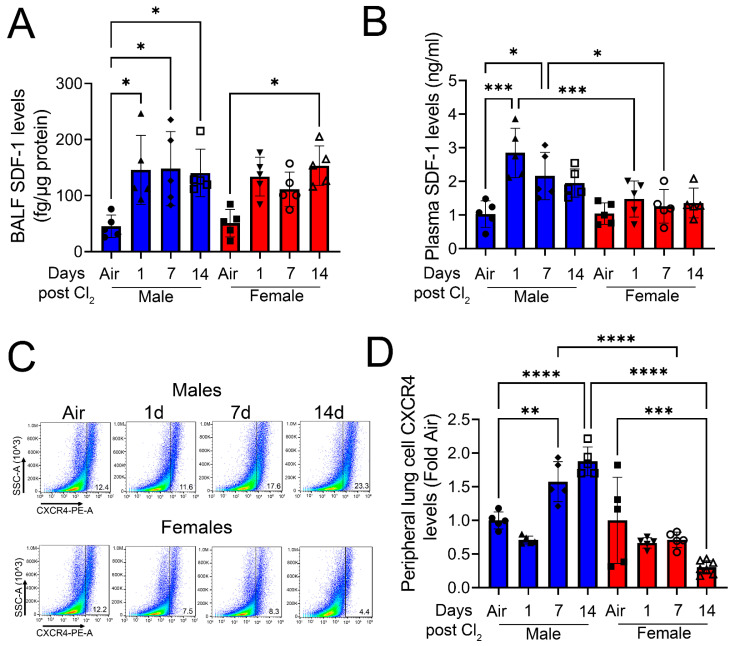
Cl_2_ increases SDF-1 and CXCR4 levels primarily in male mice. Exposure of adult C57BL/6 mice to Cl_2_ gas (500 ppm, 15 min) increases BALF SDF-1 levels in male mice on days 1, 7, and 14, but only on day 14 in female mice (**A**). Plasma SDF-1 levels are elevated in male but not female mice at 1- and 7- days post-exposure (**B**). FACS analysis (**C**) of lung cells isolated from male mice shows significantly higher peripheral levels of CXCR4 compared to their female counterparts (**D**) (n = 5–7). Individual values and means ± SEM. * *p* < 0.05, ** *p* < 0.01, *** *p* < 0.001, **** *p* < 0.0001 vs. groups at the end of individual lines; two-way ANOVA followed by Tukey’s post-hoc testing.

**Figure 4 cells-12-01719-f004:**
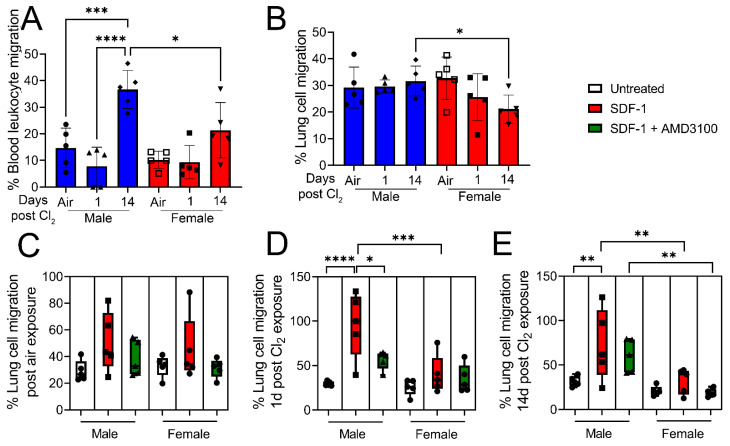
Cl_2_ increases migration of lung leukocytes in response to SDF-1 in male mice. Under basal conditions, migration of blood (**A**) and lung (**B**) leukocytes is higher in male vs. female mice 14 days post-Cl_2_ (500 ppm, 15 min) exposure. Then SDF-1 (100 ng/mL) was added to media in the lower chamber, while some lung cells were treated with AMD3100 (100 nM) overnight and migration of lung leukocytes was measured next morning. SDF-1 does not alter lung leukocyte migration in air-exposed animals (**C**). However, SDF-1 increases migration (three-fold) in leukocytes obtained from male mice 1-day post-Cl_2_ exposure (**D**), which is attenuated with AMD3100 treatment. SDF-1 also increases leukocyte migration by two-fold in cells obtained from male mice 14 days post-Cl_2_ exposure (**E**). However, there is no significant increase in migration of leukocytes obtained from female mice (n = 5). Individual values and means ± SEM. * *p* < 0.05, ** *p* < 0.01, *** *p* < 0.001, **** *p* < 0.0001 vs. groups at the end of individual lines; two-way ANOVA followed by Tukey’s post-hoc testing.

**Figure 5 cells-12-01719-f005:**
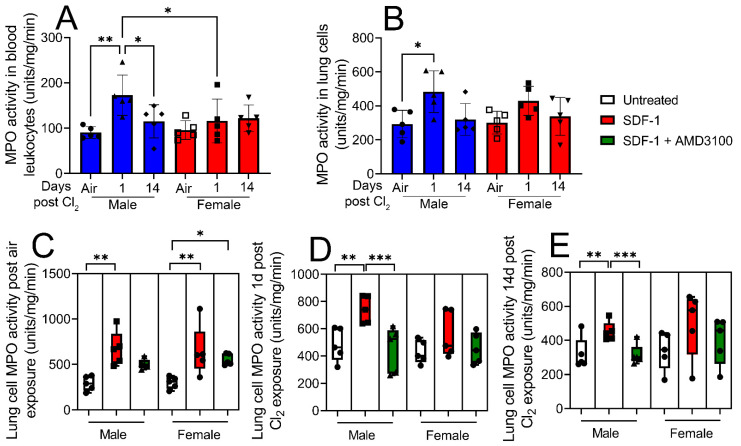
Lung leukocyte myeloperoxidase (MPO) activity increases more in response to SDF-1 in male vs. female mice post-Cl_2_ exposure. Leukocyte MPO activity was assessed in isolated blood leukocytes and lung cells from air-exposed and Cl_2_ (500 ppm, 15 min)-exposed mice, 1- and 14- days post-exposure. Blood and (**A**) lung leukocyte (**B**) MPO activity is elevated in male mice but not in female mice 1-day post-Cl_2_ exposure. Lung cells were then treated with SDF-1 (100 ng/mL) in the presence or absence of AMD3100 (100 nM) and leukocyte MPO activity assessed 1-day post-treatment. SDF-1 increases MPO activity in air-exposed male and female mice (**C**). SDF-1 increases MPO activity in lung cells obtained from male mice 1- (**D**) and 14- (**E**) days post-Cl_2_ exposure, which is attenuated by AMD3100. There is no change in MPO activity in cells obtained from female mice post-Cl_2_ exposure (n = 5). Individual values and means ± SEM. * *p* < 0.05, ** *p* < 0.01, *** *p* < 0.001, vs. groups at the end of individual lines; two-way ANOVA followed by Tukey’s post-hoc testing.

**Figure 6 cells-12-01719-f006:**
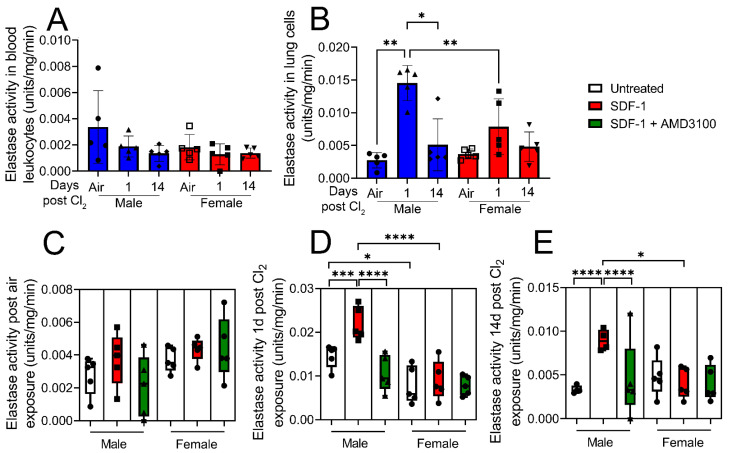
Cl_2_ exposure increases elastase activity of lung leukocytes in response to SDF-1 in male mice. Leukocyte elastase activity was assessed in isolated blood leukocytes and lung cells from air-exposed mice and from mice exposed to Cl_2_ (500 ppm, 15 min), 1- and 14- days post-exposure. Elastase activity is not changed in blood leukocytes post-Cl_2_ exposure (**A**). However, lung leukocyte elastase activity in male mice is significantly elevated 1- day post-Cl_2_ exposure. Lung cells were then treated with SDF-1 (100 ng/mL) in the presence or absence of AMD3100 (100 nM) and leukocyte elastase activity assessed 1-day post-treatment (**B**). SDF-1 does not increase elastase activity in lung cells of air-exposed mice (**C**) but increases the enzymatic activity in cells isolated from male mice 1- (**D**) and 14- (**E**) days post-Cl_2_ exposure. AMD3100 attenuates this increase. SDF-1 does not significantly alter elastase activity in cells isolated from female mice post-Cl_2_ exposure (n = 5). Individual values and means ± SEM. * *p* < 0.05, ** *p* < 0.01, *** *p* < 0.001, **** *p* < 0.0001 vs. groups at the end of individual lines; two-way ANOVA followed by Tukey’s post-hoc testing.

## Data Availability

Not applicable.

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
