# Peer review of "Sex-Based Disparities in Leukocyte Migration and Activation in Response to Inhalation Lung Injury: Role of SDF-1/CXCR4 Signaling"

_cells, 2023, doi:10.3390/cells12131719_

Round 1

Reviewer 1 Report

In their manuscript, Chatterjee et al report that CL2 exposure results in more severe injury in male compared to female mice. They investigated the activity of the enzymes MPO and elastase and expression of CXCR4 and its ligand CXCL12 or SDF-1.

Although the manuscript presents interesting results, some aspects need further clarification:

1. Fig. 1B, why 14 days is not included for the BALF cell count? Also, it seems lungs start healing after 7-14 days (less protein and less cells) while the mortality starts increasing at that moment (Fig. 1C). Can the authors explain this?

2. Authors quantify CXCL12 in plasma. However, if leukocytes migrate to the lungs, the concentration of CXCL12 in the lungs is more important since chemokines need to create a concentration gradient towards the inflamed site. What was the concentration of CXCL12 in BAL fluid in male and female mice? Could CXCL12 be detected in lung tissue extracts?

3. If CXCL12 was elevated in plasma, does this mean that also leukocyte (or neutrophil) numbers/concentrations were also higher in blood of male mice?

4. In plasma the authors already show elevated SDF-1 in male vs female mice (Fig. 3A), and they (similarly to the isolated lung cells) show elevated MPO activity in blood leukocytes in male mice (Fig. 5A). Can the authors prove that the elevated plasma SDF-1 is responsible for the elevated MPO activity by doing an ex vivo stimulation experiment with SDF-1 (similar to what they did with lung cells)?

5. Fig. 4B and line 191: The migration of leukocytes was measured after 24h. This is a very long period for an in vitro cell migration assay and may mask indirect effects. Why 24 h? Wouldn't it be more logical to do a migration experiment with blood leukocytes instead of lung leukocytes since blood leukocytes still have to migrate to the lungs while lung leukocytes already are at their end destination and the receptors involved in the migration may have been desensitized/internalized.

6. Fig. 5C; What may explain an increase in MPO activity of lung leukocytes after stimulation with SDF-1 with only air exposure (Fig.5C), in contrast to elastase (Fig. 6C)?

7. Do the authors have an explanation why the elastase activity is not increased in blood leukocytes of male mice in contrast to the MPO activity (Fig.6A) since both proteases are present in the azurophilic granules of neutrophils?

8. Do the authors think the detected CXCR4-positive lung cells are mostly neutrophils? Can they show this by flow cytometry using other neutrophil markers?

Minor comments:

1. According to international agreements, use at least once CXCL12 as systematic nomenclature for SDF-1 in the abstract.

2. Fig. 4: Indicate the dose and duration of CL2 treatment in the figure legend.

3. How many leukocytes were used for the in vitro stimulations?

4. Add the company for the Quantikine CXCL12 ELISA

5. correct on line 86: “single cell” and on line 104 “with anti CXCR4(PE)”

Reviewer 2 Report

This study investigated the effects of chlorine gas exposure on the immune response to inhalation lung injury. The authors used a mouse model to measure various lung injuries and inflammation parameters. They found that male mice were more susceptible to chlorine-induced lung injury and mortality than female mice. The authors suggested blocking CXCR4 might be a potential therapeutic strategy to reduce inflammatory lung injury in males.

The study is a shorter but focused one. The manuscript is concise and neatly written. I have a few recommendations to improve the manuscript:

1. Line 148: Please use the complete form of BALF for the first time it is used in the result section.

2. Lines 171-172: Replace "CXCR4 expression" with "CXCR4 levels," as the FACS experiment determines the protein levels, not the expression itself.

3. Figures 5 and 6: Move the legends for different treatment groups close to the data. As is, it is confusing at first glance.

4. It would be very informative if the authors investigated the extent of injury and recovery by performing H&E staining and Trichrome staining.

5. The authors focused their study on the infiltrated immune cells. The infiltration largely depends on the function of the endothelial lining of the blood vessels. The authors should investigate the differential impact of chlorine gas injury and subsequent recovery on the lung endothelial cells between males and females. I assume the endothelial cells will have a role behind the phenotype of the differential presence of the immune cells in the lung post-injury between males and females. Some potential experiments could be to see the activation state of the endothelial cells, differential gene expression, leukocyte trans-endothelial migration, etc.

6. The authors don't discuss the data enough in the results section; instead, they go about the matter in a mere commentary style. They should provide some interpretation of the data they are describing to build a complete story as they go through the results section.

Round 2

Reviewer 1 Report

Most comments have been answered.

Related to my previous comment 3: AMD3100 is a CXCR4 antagonist and inhibits retention of cells in bone marrow due to high CXCL12 expression locally in the bone marrow. However, if CXCL12 increases in plasma post Cl2 exposure, a gradient may be created from bone marrow to blood inducing leukocyte mobilization from bone marrow (similar to the attraction of CXCR4+ cells to the lung towards high CXCL12 in BAL fluid). No differences in CXCL12 levels (new fig. 3A) between male and female mice are detected at the level of the BAL fluid and as such CXCL12 levels in BAL cannot explain the higher cell numbers. However, a higher number of circulating CXCR4 positive cells or higher MFI on the CXCR4 positive cells in blood could explain the higher number of CXCR4+ cells migrating to the lung. This is why such a quantification in blood would be highly informative for the conclusions in this manuscript. If an additional experiments is needed to count these cells, this could be combined by flow cytometric characterization of the CXCR4+ cells in blood (are these mainly leukocytes or for instance endothelial/epithelial progenitor cells).

Author Response

Dear reviewer, 

You have raised excellent points about the migration of blood leukocytes to lung. We would like to explain our point of view on CXCL12/CXCR4 mediated migration. 

  1. We believe that high levels of CXCR4 in lung cells (epithelial, endothelial, immune) is a result of chlorine inhalation-induced hypoxia. Hypoxia-increases HIF-1 levels which is known to upregulate CXCR4 expression (PMID:19212630; 25444927; 17075581; 24618817). Further increased levels of cytokines in lung post chlorine inhalation also upregulate CXCR4 expression in lung cells. The overall contribution of CXCR4 positive blood leukocytes in total CXCR4 levels in lung will not be very high.
  2. Moreover, a recent study found that the surface CXCR4 expression increased in extravascular, but not intravascular, neutrophils in the lungs of LPS-induced lung injury model mice (PMID: 21460863). 
  3. Therefore, the surface expression of CXCR4 on blood leukocytes would be lower than CXCR4 expression on lung cells post chlorine gas exposure, which would create a gradient for blood neutrophils to migrate towards the injured tissue with higher CXCL12/CXCR4 expression/signaling.

Reviewer 2 Report

The authors have reasonably improved the manuscript.

Author Response

Thank you for your effort and guidance.